# The Prognostic Nutritional Index and Nutritional Risk Index Are Associated with Disease Activity in Patients with Systemic Lupus Erythematosus

**DOI:** 10.3390/nu11030638

**Published:** 2019-03-16

**Authors:** María Correa-Rodríguez, Gabriela Pocovi-Gerardino, José-Luis Callejas-Rubio, Raquel Ríos Fernández, María Martín-Amada, María-Gracia Cruz-Caparros, Norberto Ortego-Centeno, Blanca Rueda-Medina

**Affiliations:** 1Department of Nursing, Health Sciences Faculty, University of Granada (UGR), Avenida de la Ilustración s/n, 18100-Armilla (Granada), Spain; macoro@ugr.es (M.C.-R.); blarume@ugr.es (B.R.-M.); 2Health Sciences Facultuy. PhD student of the Public Health and Clinic Medicine program of the University of Granada (UGR), Avenida de la Ilustración s/n, 18100-Armilla (Granada), Spain; 3Instituto de Investigación Biosanitaria, IBS. Avda. de Madrid, 15. Pabellón de consultas externas 2, 2ª planta, 18012 Granada, Spain; 4Unidad de Enfermedades Autoinmunes Sistémicas, Servicio de Medicina Interna, Hospital Universitario San Cecilio, Av. de la Investigación, s/n, 18016 Granada, Spain; jlcalleja@telefonica.net (J.-L.C.-R.); rriosfer@hotmail.com (R.R.F.); nortego@gmail.com (N.O.-C.); 5Unidad de Enfermedades Autoinmunes Sistémicas, Servicio de Medicina Interna, Complejo Hospitalario de Jaén, Av. del Ejército Español, 10, 23007 Jaén, Spain; madfmar@gmail.com; 6Unidad de Enfermedades Autoinmunes Sistémicas, Servicio de Medicina Interna, Hospital de Poniente, Carretera de Almerimar, 31, 04700 El Ejido, Almería, Spain; grcrca@hotmail.com

**Keywords:** prognostic nutritional index, controlling nutritional status, nutritional risk index, systemic lupus erythematosus, lupus disease activity, lupus damage

## Abstract

The prognostic nutritional index (PNI), controlling nutritional status (CONUT) score and nutritional risk index (NRI) have been described as useful screening tools for patient prognosis in several diseases. The aim of this study was to examine the relationship between PNI, CONUT and NRI with clinical disease activity and damage in 173 patients with systemic lupus erythematous (SLE). Disease activity was assessed with the SLE disease activity index (SLEDAI-2K), and disease-related organ damage was assessed using the SLICC/ACR damage index (SDI) damage index. PNI and NRI were significantly lower in active SLE patients than in inactive SLE patients (*p* < 0.001 and *p* = 0.012, respectively). PNI was inversely correlated with the SLEDAI score (*p* < 0.001) and NRI positively correlated with SLEDAI and SDI scores (*p* = 0.027 and *p* < 0.001). Linear regression analysis adjusting for age, sex and medications showed that PNI was inversely correlated with SLEDAI (β (95% CI) = −0.176 (−0.254, −0.098), *p* < 0.001) and NRI positively correlated with SLEDAI (β (95% CI) = 0.056 (0.019, 0.093), *p* = 0.003) and SDI (β (95% CI) = 0.047 (0.031, 0.063), *p* < 0.001). PNI (odds ratio (OR) 0.884, 95% confidence interval (CI) 0.809–0.967, *p* = 0.007) and NRI ((OR) 1.067, 95% CI 1.028–1.108, *p* = 0.001) were independent predictors of active SLE. These findings suggest that PNI and NRI may be useful markers to identify active SLE in clinical practice.

## 1. Introduction

Systemic lupus erythematosus (SLE) is a heterogeneous, chronic, inflammatory autoimmune disease that involves multiple organ systems and displays a variable clinical score [1]. The precise aetiology mechanisms are unknown, but SLE seems to be the result of the interaction between genetic, hormonal, environmental and immune abnormalities [2]. The heterogeneity of clinical manifestations of SLE complicate the management of these patients in clinical practice. Therefore, there is still a need for novel, clinically useful and easily assessed biomarkers that can identify SLE disease activity.

Nutritional status has been widely correlated with immunity, where undernutrition is associated with immunosuppression and increased susceptibility to infection, whereas overnutrition and/or overeating is associated with low-grade chronic inflammation increasing the risk and affecting the prognosis in metabolic, cardiovascular and autoimmune diseases [3]. In general, SLE patients are undergoing multiple medical treatments, and can go through chronic fatigue, depression and the presence of comorbidities which lead to changes in appetite and an increased risk of malnutrition [4]. Serum albumin is a well-known negative acute-phase protein whose levels decrease during inflammation and malnutrition [5,6]. Inflammation may reduce albumin concentration by decreasing its rate of synthesis, and it has also been associated with a greater fractional catabolic rate [6]. Subnormal levels have commonly been reported in SLE patients [7,8] and serum albumin has previously been established as a potential surrogate marker of SLE disease activity [8,9,10].

On the other hand, lymphopaenia is one of the most frequent clinical manifestations in SLE, reported in up to 93% of patients [11]. Traditionally, and according to the American College of Rheumatology (ACR), lymphopaenia only offers limited clinical usefulness as one of the haematologic criteria for SLE classification [12]. However, previous studies have demonstrated that lymphopaenia has a significant clinical value because it can be associated with disease activity and damage accrual in SLE patients, proposing that lymphopaenia may be an expression of disease activity [13,14,15].

Some authors have reported that certain objective assessment indexes reflecting the immune-nutritional status of patients, such as the prognostic nutritional index (PNI) [16], the controlling nutritional status (CONUT) score [17], and the nutritional risk index (NRI) [18,19], which are calculated using serum albumin level and lymphocyte count, represent useful screening tools for patient prognosis in several diseases [20,21,22,23,24,25]. PNI was at first described as a simple and objective indicator of postoperative outcomes in patients undergoing cancer surgery [16,26]. CONUT, which uses, besides albumin concentration and lymphocyte count, total cholesterol (TC) concentration, was developed as a screening tool for the early detection of undernutrition and poor nutritional status [17]. NRI, developed at first for assessing patients with total parenteral nutrition, was found to be a sensitive and specific predictor for identifying patients with a risk of complications after surgery [18] and later adapted and validated in the geriatric population [19]. Interestingly, Ahn et al. [27] recently showed that PNI correlated with SLE disease activity, claiming that PNI may provide an effective estimation of SLE activity. However, to the best of our knowledge, there have been no studies into the significance of other immune-nutritional indexes, including the CONUT score and NRI, in terms of predicting disease activity and damage accrual in SLE patients.

Predicting active SLE using reliable markers is of particular interest when it comes to implementing useful preventive strategies in clinical practice. Additionally, since PNI, CONUT and NRI scores can be obtained from laboratory data using blood samples, clinicians would be able to objectively, simply and continuously evaluate the immune-nutritional status of SLE patients. Considering previous evidence suggests that markers of serum albumin level and lymphocyte count might be able to predict active SLE and disease prognosis, we hypothesised that PNI, CONUT and NRI could represent convenient and cost-effective biomarkers for predicting disease activity and damage accrual in lupus patients. Therefore, the objective of this study was to assess nutritional status and risk through PNI, CONUT and NRI according to SLE activity and the damage accrual of SLE.

## 2. Materials and Methods

### 2.1. Study Population

A cross-sectional study was conducted among 173 subjects diagnosed with SLE attending the Outpatient Clinic of the Systemic Autoimmune Diseases Unit across three public hospitals in the Andalusian region of Spain from January 2016 to December 2018. All patients met the SLE revised diagnostic criteria of the ACR or Systemic Lupus International Collaborating Clinics (SLICC) classificatory criteria [28]. According to these, a person is classified as having lupus if he or she has “lupus nephritis” (the presence of antinuclear antibodies or anti-double-stranded DNA antibodies) or meets four criteria (with at least one criterion being clinical and at least one criterion being immunological) from a series of clinical and analytical manifestations characteristic of the disease [28]. The participants had been confirmed as suffering SLE at least one year prior to the study and were clinically stable, with no changes in the systemic lupus erythematosus disease activity index (SLEDAI) [29] and/or medical treatment over the six months immediately prior to the study. We considered as exclusion criteria patients with cerebrovascular disease, stablished ischemic heart diseases, serum creatinine ≥ 1.5 mg/dL, active infections, major trauma or surgery in the previous six months, pregnancy and/or other chronic and/or autoimmune systemic conditions (i.e., rheumatoid arthritis, cancer, multiple sclerosis, type 1 diabetes) not related with the main disease. Written informed consent was obtained from each participant and the study was approved by local ethics committees and conducted in accordance with the Declaration of Helsinki.

### 2.2. Clinical Disease Activity and Damage Accrual

The activity of the disease was assessed with the (SLEDAI-2K), a well-established and accepted disease activity score [29]. SLEDAI is a list of 24 components, 16 of which are clinicals and eight are laboratory results. These components are scored based on whether these manifestations are present or absent in the previous 10 days. The total score of SLEDAI-2 K is the sum of all 24 descriptor scores and falls between 0 and 105, and a score of 6 is considered clinically important. SLEDAI-2K is a modification of SLEDAI that allows the documentation of ongoing disease activity in some clinical components as skin rash, alopecia, mucosal ulcers and proteinuria as opposed to only new occurrences as defined in the original SLEDAI [29]. Meaningful improvement is best defined as a reduction in SLEDAI-2 K of 4 [30]. Active SLE was defined as a SLEDAI-2K value of ≥ 5 [31].

Disease-related organ damage was assessed by using the SLICC/ACR damage index (SDI) [32]. This instrument has been developed to assess irreversible damage in SLE patients, independently of its cause [33]. The maximum possible score is 47. The SDI damage score gradually increases over time, and patients with higher damage scores early in the course of disease have been associated with poor prognosis and increased mortality [34].

### 2.3. Laboratory Measurements

Venous blood samples were collected between 07:30 and 10:00 AM after an overnight fast and then centrifuged for 15 min to obtain serum. The serum was analyzed immediately to obtain the biochemical variables determined by standard laboratory methods. Anti-double stranded DNA (Anti-dsDNA) antibodies were measured using a commercially available BioPlex 2200 System (Bio-Rad, Hercules, CA, USA), which is an automated analyser that detects antibodies for several antigens in one tube. Results are expressed in IU/mL, and the cut-off values established by the manufacturer are 5–9 IU/mL (indeterminate) and ≥ 10 IU/mL (positive). Human complement components C3 and C4 and high sensitivity C-reactive protein (hsCRP) levels were determined quantitatively in serum samples by immunoturbidimetric assay (Beckman Coulter AU System CRP Latex reagent) in a Beckman Coulter analyser (AU5800 Analyzer, Beckman Coulter, CA, USA). Normal ranges are as follows: 90–180 mg/dL for C3, 10–40 mg/dL for C4 and 0.2–80 mg/L (1.9–761.9 nmol/L) for hsCRP. Laboratory testing in a clinical routine including total cholesterol, serum albumin, lymphocyte count, white blood cell (WBC) count and platelet count was performed at Clinical Chemistry Laboratories at the San Cecilio Hospital (Granada, Spain).

### 2.4. Calculation of PNI, CONUT and NRI Scores

PNI was calculated according to the following formula: 10 x serum albumin value (g/dL) + 0.005 × peripheral lymphocyte count (/mm^3^) [26]. A higher PNI indicates a high risk of malnutrition [26]. The CONUT score is based on a calculation using the serum albumin level, total lymphocyte count and total cholesterol level (range 0–12, higher = worse) [17]. In this scoring system, points are assigned according to different ranges for the laboratory measures: serum albumin ≥ 3.5 g/dL: 0 points; 3.49–3.0: 2 points; 2.99–2.5: 4 points; and < 2.5: 6 points; lymphocytes ≥1600/μL: 0 points; 1200–1599: 1 point; 800–1199: 2 points; and < 800: 3 points; and total cholesterol ≥ 180 mg/dL: 0 points; 140–179: 1 point; 100–139: 2 points; and < 100: 3 points. The nutritional risk index (NRI) [18] was calculated as NRI = (1.519 × serum albumin, g/dL) + (41.7 × weight (kg)/ideal body weight (IBW; kg)) [19]. Ideal body weight was calculated using the Lorentz formulae; i.e., height (cm) − 100 − ((height (cm) – 150)/4) for men and height (cm) − 100 − ((height (cm) – 150)/2.5) for women. Body mass index (BMI) was calculated and classified according to the World Health Organisation (WHO) [35]. A lower NRI indicates a higher risk of malnutrition [18].

### 2.5. Statistical Analysis

SPSS^®^ Statistics version 21.0 (IBM, Chicago, IL, USA) was used for all analyses. Continuous variables were reported as mean ± standard deviation and categorical variables as frequencies and percentages (n and %). The Kolmogorov–Smirnov test was used to verify data distribution normality. The Kolmogorov–Smirnov normality test was conducted before association analysis. Data were distributed in two groups according to SLE activity (active SLE (SLEDAI ≥ 5) and inactive SLE (SLEDAI < 5)) based on the SLEDAI. To compare the two groups, we used the Mann–Whitney U test and Student’s t-test for continuous data and χ^2^ for categorical data. Data were expressed as mean ± standard deviation or median (interquartile range) for continuous variables and as frequencies for categorical variables. Due to their skewed distribution, the following variables were log-transformed before analysis: anti-dsDNA titres, complement C3 and C4, WBC count, platelets count, hsCRP, and SLEDAI and SDI scores. To aid interpretation, data were back-transformed from the log scale for presentation in the results. Pearson’s correlation analysis was used to elucidate the correlation between the immune-nutritional indexes and clinical disease variables. Linear regression analyses were used to examine the relationships between SLEDAI and SDI scores, laboratory variables, and PNI, CONUT and NRI scores. Age, sex and medical treatments—immunosuppressants (mycophenolate mofetil, azathioprine, methotrexate), antimalarials and/or corticosteroids—were considered confounding factors and adjusted for in the prior analysis. Logistic regression models were used to estimate odds ratios for active SLE after adjusting for age, sex and medical treatment. The variance inflation factor (VIF) was used to diagnose multicollinearity. *p* values of < 0.05 were taken as statistically significant.

## 3. Results

The main characteristics of the study subjects are shown in Table 1, for the full sample, and stratified by active and inactive SLE based on the SLEDAI.

A total of 41 patients (23.7%) were classified as having active SLE and 132 (76.3%) as having inactive SLE. In 41 active SLE patients, eight (20.5%) had renal affectation or lupus nephritis, 22 (53.7%) had arthritis, 31 (75,6%) had oral ulcers, 34 (87.2%) had leuco-lymphopenia, six (14.6%) had thrombocytopenia, five (12.12%) had neurological affectation and three (7.3%) had pericarditis (data not shown). Most patients enrolled in the study were females (89%) and the mean age of the population was 46.6 ± 13.0 years. The mean time since SLE diagnosis was 8.52 (1–37) years. According to their medical records, 83.7% of patients were taking antimalarials, 40.7% immunosuppressants, 43.6% corticosteroids (mean dose 3.05 mg) and 25.9% statins. Significant differences in immunosuppressant and corticoid use were observed between patients with active and inactive SLE (*p* < 0.001 and *p* = 0.017, respectively). We also found significant differences between patients with active and inactive SLE in weight, BMI, serum albumin concentration, anti-dsDNA, complement C4 level and hsCRP. As expected, patients with active SLE had significantly higher SLEDAI and SDI scores than patients with inactive SLE (7.00 ± 1.67 vs. 1.50 ± 1.55, *p* < 0.001 and 1.92 ± 1.27 vs. 0.85 ± 1.20, *p* < 0.001, respectively). Regarding immuno-nutritional indexes, PNI and NRI were significantly lower in active SLE patients than in inactive SLE patients (46.27 ± 4.35 vs. 49.57 ± 5.14, *p* < 0.001 and 61.04 ± 12.87 vs. 55.35 ± 9.85, *p* = 0.012, respectively).

Pearson’s correlation analysis revealed that PNI was negatively correlated with the SLEDAI score (*p* < 0.001) and NRI positively correlated with SLEDAI and SDI scores (*p* = 0.027 and *p* < 0.001). Also, hsCRP was positively correlated with SLEDAI and SDI scores (*p* = 0.004 and *p* < 0.001, respectively) (data not shown).

This was also observed in linear regression analysis where hsCRP was significantly correlated with SLEDAI and SDI scores after adjusting for age, sex and medical treatment (Table 2). Anti-dsDNA and complement C3 were correlated with SDI while complement C4 and WBC count were only correlated with SLEDAI after adjustments for covariates. PNI was inversely correlated with SLEDAI [β (95% CI) = −0.176 (−0.254, −0.098), *p* < 0.001] and NRI positively correlated with SLEDAI (β (95% CI) = 0.056 (0.019, 0.093), *p* = 0.003) and SDI scores (β (95% CI) = 0.047(0.031, 0.063), *p* < 0.001).

Logistic regression analysis revealed that hsCRP (odds ratio (OR) 1.104, 95% CI 1.017–1.198, *p* = 0.018), complement C4 (OR 0.959, 95% CI 0.920–0.999, *p* = 0.046), PNI (OR 0.884, 95% CI 0.809–0.967, *p* = 0.007) and NRI (OR 1.067, 95% CI 1.028–1.108, *p* = 0.001) were independent predictors of active SLE after adjusting for age, sex and medical treatment (Table 3).

## 4. Discussion

Our study demonstrates that patients with active SLE had lower values of PNI and NRI than those with active SLE. More importantly, we identified a relationship between NRI and damage accrual measured by SDI, and both PNI and NRI correlated with disease activity measured by SLEDAI in SLE patients. In addition, we found that PNI and NRI were independent predictors of active SLE.

As far as we are aware, only one study has investigated the association of PNI with disease activity in SLE [27]. Our findings agree with those reported by Ahn et al., where PNI was an independent predictor of active SLE in a population of Korean patients [27]. Thus, the present study considered together with previous work suggests that PNI may be a useful index for the evaluation of disease activity in lupus patients. However, the relationship between other well-known immuno-nutritional indexes, such as the CONUT score and NRI, and clinical disease activity have not been investigated previously in SLE. Both PNI and NRI are estimated based on serum albumin levels and lymphocyte count. Previous studies have investigated these two parameters independently to evaluate their relationship with SLE activity. Serum albumin, a readily-available, routine measurement in SLE, has shown a negative association with disease activity in lupus patients [8,9,10]. Recently, Idborg et al. proposed the use of plasma albumin as a potential biomarker of disease activity in SLE [8], while Yip et al. showed that higher SLEDAI scores correlated with lower serum albumin levels in a large population of patients [9]. This may be explained by the fact that reduced albumin levels in active patients are secondary to kidney damage [36]. Additionally, the acute-phase response may affect albumin concentrations [37]. In the same vein, lymphopaenia is a well-known manifestation in SLE [11]. Previous publications have demonstrated that lymphopaenia correlated with SLE disease activity [13,14,15]. In a multi-ethnic study, Vilá et al. reported that lymphopaenia is associated with higher disease activity and damage accrual [13], and in a prospective study, after one year of follow-up, the SLEDAI score was shown to be predicted by lymphopaenia [14]. Similarly, lymphopaenia was related to disease activity and organ damage in a paediatric population with SLE [15].

Thus, taking our findings into account along with the above-mentioned studies, we can conclude that PNI and NRI, calculated using both serum albumin and lymphocyte count, might be useful in clinical practice as straightforward, inexpensive biomarkers for monitoring disease activity in SLE patients.

There were potential limitations to this study. Firstly, due to its cross-sectional design, we cannot draw any causal conclusions. Longitudinal studies are required to evaluate how PNI and NRI influence disease activity and damage accrual in lupus patients. Secondly, although this study was conducted in a well-characterised lupus population, with the inclusion of low-stage disease patients and exclusion of patients with other clinical conditions, given it is impossible to discontinue treatment strategies for a period prior to sampling, it can be argued that lymphopaenia or altered serum albumin levels could be caused by immunosuppressive agents rather than lupus. However, this does not seem to be a determining factor in this study as our results remained significant after adjusting for medical treatment. Also, since we only included European Caucasian patients, these findings cannot be generalised to other ethnicities. Thus, racial and ethnic differences in the relationship between PNI and NRI scores and SLE activity require further investigation. In contrast, to the best of our knowledge, this study was the first to address the relationship between NRI and disease activity and damage accrual in lupus patients. Future prospective studies are needed to identify its clinical significance in lupus activity.

## 5. Conclusions

In conclusion, NRI is associated with increased disease activity and organ damage on SLE and appears to be an immuno-nutritional index in the evaluation of SLE activity and damage accrual. Moreover, our results confirm the association between PNI and disease activity in SLE patients. We are aware that the results reported are preliminary and further studies on a larger sample are required.

## Figures and Tables

**Table 1 nutrients-11-00638-t001:** Descriptive characteristics of patients with active and inactive systemic lupus erythematous (SLE).

Characteristics	Total (*n* = 172)	Active SLE ^b^ (*n* = 41)	Inactive SLE ^b^ (*n* = 132)	*p* Value
Female	154 (89.0)	39 (95.1)	115 (87.1)	0.152
Age (years)	46.6 ± 13.07	44.05 ± 12.37	47.41 ± 13.22	0.140
Height (m)	1.61 ± 0.08	1.59 ± 0.07	1.61 ± 0.08	0.255
Weight (kg)	68.06 ± 15.79	73.19 ± 18.74	66.46 ± 14.47	0.039
BMI (kg/m^2^)	26.03 ± 5.81	28.54 ± 6.90	25.33 ± 5.29	0.016
Antimalarial use	144 (83.7)	35 (87.5)	109 (82.6)	0.460
Immunosuppressor ^a^ use	70 (40.7)	26 (65.0)	44 (33.3)	<0.001
Corticoid use	75 (43.6)	24 (60.0)	51 (38.6)	0.017
Total cholesterol (mg/dL)	181.6 ± 39.6	178.9 ± 34.7	182.4 ± 41.1	0.582
Albumin (g/dL)	4.0 ± 0.3	3.8 ± 0.3	4.1 ± 0.3	<0.001
Lymphocyte count (μL)	1576.3 ± 682.3	1465.0 ± 516.7	1610.8 ± 724.3	0.158
Anti-dsDNA (IU/mL)	18.9 (0–300.0)	35.2 (0–174.0)	13.7 (0–300.0)	0.011
Complement C3 level (mg/dL)	107.1 (11.7–199.0)	102.1 (37.1–99.0)	108.7 (11.7–195.0)	0.254
Complement C4 level (mg/dL)	21.5 (0.8–114.4)	17.1 (0.8–45.2)	22.9 (3.7–114.4)	0.003
WBC count (× 1000/μL)	5.7 (0–4.7)	5.5 (3.4–11.9)	5.7 (0–14.7)	0.436
Platelet count (× 1000/(μL)	222.3 (0.2–502.0)	238.2 (95.0–502.0)	217.3 (0.2–385.0)	0.175
hsCRP (mg/dL)	3.69 (0.20–24.10)	5.46 (0.20–24.10)	3.14 (0.20–19.50)	0.022
SLEDAI score	2.8 (0–12)	7.0 ± 1.6	1.5 ± 1.5	<0.001
SDI score	1.1 (0–9)	1.9 ± 1.2	0.8 ± 1.2	<0.001
PNI	48.7 ± 5.1	46.2 ± 4.3	49.5 ± 5.1	<0.001
CONUT	1.7 ± 1.5	1.9 ± 1.4	1.6 ± 1.5	0.283
NRI	56.7 ± 10.8	61.0 ± 12.8	55.3 ± 9.8	0.012

^a^ Immunosuppressor: mycophenolate mofetil, azathioprine or methotrexate. ^b^ Active SLE (SLEDAI ≥ 5); Inactive SLE (SLEDAI < 5). Mann–Whitney U test and Student’s *t*-test for continuous data and χ^2^ for categorical data were used to compare between groups. BMI, body mass index; Anti-dsDNA, Anti-double stranded DNA antibodies; WBC, white blood cells; hs-CRP, high-sensitivity C-reactive protein; SLEDAI, systemic lupus erythematosus disease activity index; SDI, damage index for systemic lupus erythematosus; PNI, prognostic nutritional index; CONUT, controlling nutritional status; NRI, nutritional risk index.

**Table 2 nutrients-11-00638-t002:** Linear regression analysis between clinical disease activity variables and immuno-nutritional indexes and SLEDAI and SDI scores.

	SLEDAI Score			SDI Score		
	*β*	95% CI	*p* Value	*β*	95% CI	*p* Value
hsCRP (mg/dL)	0.116	0.023, 0.209	0.015	0.083	0.041, 0.124	<0.001
Anti-dsDNA (IU/mL)	−1.201	−2.467, 0.065	0.063	−0.712	−1.302, −0.121	0.018
Complement C3 level (mg/dL)	−0.008	−0.023, 0.006	0.264	0.010	0.004, 0.017	0.002
Complement C4 level (mg/dL)	−0.039	−0.070, −0.007	0.017	0.009	−0.015, 0.015	0.963
WBC count (μL)	−0.197	−0.394, −0.001	0.049	0.036	−0.055, 0.128	0.432
Platelet count (×1000/(μL)	0.000	−0.006, 0.005	0.981	−0.001	−0.004, 0.001	0.241
PNI	−0.176	−0.254, −0.098	<0.001	−0.020	−0.058, 0.018	0.307
CONUT	0.168	−0.100, 0.436	0.218	0.013	0.111, 0.137	0.834
NRI	0.056	0.019, 0.093	0.003	0.047	0.031, 0.063	<0.001

Adjusted for age, sex and medical treatment (immunosuppressants, antimalarials and/or corticosteroids). hs-CRP, high-sensitivity C-reactive protein; Anti-dsDNA, anti-double stranded DNA antibodies; WBC, white blood cells; PNI, prognostic nutritional index; CONUT, controlling nutritional status; NRI, nutritional risk index.

**Table 3 nutrients-11-00638-t003:** Logistic regression analysis of clinical disease activity variables and immuno-nutritional indexes in patients with active and inactive SLE.

	Odds Ratio	95% CI	*p* Value
hsCRP (mg/dL)	1.104	1.017, 1.198	0.018
Anti-dsDNA (IU/mL)	0.394	0.134, 1.155	0.090
Complement C3 level (mg/dL)	0.999	0.9826 1.013	0.933
Complement C4 level (mg/dL)	0.959	0.920, 0.999	0.046
WBC count (μL)	0.943	0.785, 1.134	0.535
Platelet count (×1000/(μL)	1.002	0.997, 1.008	0.350
PNI	0.884	0.809, 0.967	0.007
CONUT	1.106	0.867, 1.412	0.418
NRI	1.067	1.028, 1.108	0.001

Adjusted for age, sex and medical treatment (immunosuppressants, antimalarials and/or corticosteroids). hs-CRP, high-sensitivity C-reactive protein; WBC, white blood cells; PNI, prognostic nutritional index; CONUT, controlling nutritional status; NRI, nutritional risk index.

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
