# Peer review of "The Prognostic Nutritional Index and Nutritional Risk Index Are Associated with Disease Activity in Patients with Systemic Lupus Erythematosus"

_nutrients, 2019, doi:10.3390/nu11030638_

Round 1

Reviewer 1 Report

This was an interesting study and a novel one, congratulations to the authors.

Few comments for improvement:

- The reference style appears heterogenous in the text, indicating previous (?) submissions? For example L72, L76 the references are not per journal style.

The introduction needs a couple more sentences describing increased nutritional risk and disease activity in SLE.

- You need to explain what each nutritional risk index is assessing. This is missing from the manuscript. Additionally, please note that these indexes are assessing nutritional risk and not nutritional assessment. Therefore, L71 needs to be rephrased perhaps to "...certain objective nutritional risk indexes reflecting...".

L72 and L141: PNI reference is wrong! The corrected reference is:

Buzby GP, Mullen JL, Matthews DC, Hobbs CL, Rosato EF. Prognostic nutritional index in gastrointestinal surgery. Am J Surg. 1980;139(1):160-7.

L94: SLICC is not defined.

L111: change word "items" which is very frequent in that paragraph with "components".

L119: what does SDI stands for?

L119: SLICC must be defined earlier so no need for full text here.

Paragraph 2.4: You need to explain what high/low scores of each index mean. Are we aiming high or low?

L148: The NRI reference is wrong! You are citing the geriatric NRI instead! The correct reference is Perioperative total parenteral nutrition in surgical patients. The Veterans Affairs Total Parenteral Nutrition Cooperative Study Group. N Engl J Med. 1991;325:525–32.

-Why did you use Pearson's instead of regressions? It is lower quality.

L166: did you fulfill the prerequisites for logistic regression? Please list them.

-Table 1: Height is in meters and you suggest that these are cm. A sample with a mean height of 1.61 cm must be humans!

Table 1: what did you consider as immunosuppressors? This should be included in the footnote of the table. Is via D included?

Table 1: TC has no units.

Table 1: which tests were used to compare between active and inactive SLE? This should be mentioned in the footnote and additionally in the statistical analysis paragraph.

L178: "...inactive SLE based on the SLEDAI".

Table 2: All presented correlations are weak, except for SDI and NRI, which is fair. I suggest removing the table and just mentioning this one correlation in the text. Overall, why choose correlations instead of regression? Additionally, if NRI and SDI have common components, eg. albumin, there is no need to present anything with correlations. Keep the important stuff and avoid the less meaningful to add quality to your study.

Table 4: CONUT is written twice in the footnote.

- Is nutritional risk increased due to SLE activity or is increased nutritional risk leading to active SLE?

I believe that the use of nutritional risk indexes to identify active SLE is misleading given that you did not evaluate agreement between pairs of indexes. It is safer to conclude that increased nutritional risk is associated with increased disease activity and organ damage. It is my belief that this is preferred and safer as a research question and conclusion based on your analyses. So, I would also like the research question to be "to assess nutritional risk according to SLE activity, and explain active SLE via nutritional risk indexes."

References:

Ref 11 has issues.

Best regards!

Author Response

REVIEWER #1

This was an interesting study and a novel one, congratulations to the authors.

We thank the reviewer for this comment.

Few comments for improvement:

- The reference style appears heterogenous in the text, indicating previous (?) submissions? For example L72, L76 the references are not per journal style.

We apologize for these errors in referring, it was just a mistake during the edition of the manuscript. Thanks for noticing this.

-The introduction needs a couple more sentences describing increased nutritional risk and disease activity in SLE.

Thanks for your constructive comment, we have added a couple of lines about this fact in the revised version of the manuscript (see page lines 58-60).

- You need to explain what each nutritional risk index is assessing. This is missing from the manuscript. Additionally, please note that these indexes are assessing nutritional risk and not nutritional assessment. Therefore, L71 needs to be rephrased perhaps to "...certain objective nutritional risk indexes reflecting...".

We thank the reviewer for this important suggestion. We have modified the paragraph including additional information about each index and also rephrased the referred lines according to the suggestion. We have also revised the references to make sure that they match the original one of each index (see lines 77-83)

-L72 and L141: PNI reference is wrong! The corrected reference is:

Buzby GP, Mullen JL, Matthews DC, Hobbs CL, Rosato EF. Prognostic nutritional index in gastrointestinal surgery. Am J Surg. 1980;139(1):160-7.

We apologize for this mistake. As suggested, we have modified the reference for the first and original one where PNI was developed and validated (lines 74,78).

-L94: SLICC is not defined.

Thanks for this suggestion. We have specified the acronym SLICC in lines 104-105.

-L111: change word "items" which is very frequent in that paragraph with "components".

Thanks for this valuable suggestion. We have modified it through the paragraph (lines 121-128).

-L119: what does SDI stands for?

SDI is the acronym used for shortening the Systemic Lupus International Collaborating Clinics/American College of Rheumatology (SLICC/ACR) Damage Index. This information has been included in the revised version of the manuscript. It was developed to assess an ongoing reflection of disease activity in SLE patients and to measure irreversible damage resulting from SLE disease activity and its treatment. All damage is scored from the time of SLE diagnosis onward regardless of whether or not the damage is attributed to lupus.

Ghazali WSW, Daud SMM, Mohammad N, Wong KK. SLICC damage index score in systemic lupus erythematosus patients and its associated factors. Medicine (Baltimore). 2018;97(42):e12787.

-L119: SLICC must be defined earlier so no need for full text here.

As suggested, we have defined the acronym SLICC before and removed the full text in these lines (see line 130).

-Paragraph 2.4: You need to explain what high/low scores of each index mean. Are we aiming high or low?

Thanks for this suggestion. We have added a few lines regarding this fact for each index in the revised version of the manuscript (see lines 155,156, 165,166).

-L148: The NRI reference is wrong! You are citing the geriatric NRI instead! The correct reference is Perioperative total parenteral nutrition in surgical patients.

The Veterans Affairs Total Parenteral Nutrition Cooperative Study Group. N Engl J Med. 1991;325:525–32.

We apologize for this mistake. As suggested, we have modified the reference in the referred line and in the beginning of the manuscript (lines 82, 161). Also, we have added the correct reference (ref 18).  

-Why did you use Pearson's instead of regressions? It is lower quality.

Taking into account this comment and the presented by reviewer #2, Table 2 has been removed from the revised version of the manuscript. We have just mentioned the significant correlations found in the text (see lines 219-222). We agree with the reviewer that regressions are higher quality. Thus, Linear regression analysis is now showed in Table 2 of the revised manuscript.

-L166: did you fulfill the prerequisites for logistic regression? Please list them.

We appreciate this useful comment. We would like to note that, as far as we concerned, logistic regression does not require a linear relationship between the dependent and independent variables.  Also, the error terms (residuals) do not need to be normally distributed and homoscedasticity is not required.  However, we have considered the following assumptions:

- First, since logistic regression requires the observations to be independent of each other, in this study the observations should not come from repeated measurements or matched data.

- Second, taking into account that logistic regression requires there to be little or no multicollinearity among the independent variables, the variance inflation factor (VIF), which quantifies the extent of correlation between one predictor and the other predictors in a model, was used for diagnosing multicollinearity. 

-Finally, since logistic regression typically requires a large sample size, in this study we included a cohort 172 SLE patients.

-Table 1: Height is in meters and you suggest that these are cm. A sample with a mean height of 1.61 cm must be humans!

Thanks for noticing this. We have fixed this issue.

-Table 1: what did you consider as immunosuppressors? This should be included in the footnote of the table. Is via D included?

We considered as immunosupressors the use of mycophenolate mofetil, azathioprine and methotrexate. As suggested, we have included this information in footnotes (table 1) but also in the methods section (lines 184,185). Vitamin D was not included in the referred analysis.

-Table 1: TC has no units.

We have added the unit used for TC in the referred table. Thanks for the suggestion.

-Table 1: which tests were used to compare between active and inactive SLE? This should be mentioned in the footnote and additionally in the statistical analysis paragraph.

First, data were distributed in two groups Active SLE (SLEDAI ≥5) and inactive SLE (SLEDAI<5). To compare the two groups, we used the Mann-Whitney U test and Student’s t-test for continuous data and χ2 for categorical data. This information has been added in the revised version of the manuscript (see lines 173-176) and footnotes (table 1).

As suggested, this information has been added in the statistical analysis paragraph and also as a footnote in table 1.

-L178: "...inactive SLE based on the SLEDAI".

Thanks for the suggestion, we have added this phrase.

-Table 2: All presented correlations are weak, except for SDI and NRI, which is fair. I suggest removing the table and just mentioning this one correlation in the text. Overall, why choose correlations instead of regression? Additionally, if NRI and SDI have common components, eg. albumin, there is no need to present anything with correlations. Keep the important stuff and avoid the less meaningful to add quality to your study.

Thanks for this useful suggestion. According to the previous comment, we have deleted Table 2 in the revised version of the manuscript, and we have just mentioned the significant correlations in the text (see lines 219-222). Linear regression analysis is now showed in Table 2 of the revised manuscript.

-Table 4: CONUT is written twice in the footnote.

We have fixed it. Thanks for noticing this.

- Is nutritional risk increased due to SLE activity or is increased nutritional risk leading to active SLE?

I believe that the use of nutritional risk indexes to identify active SLE is misleading given that you did not evaluate agreement between pairs of indexes. It is safer to conclude that increased nutritional risk is associated with increased disease activity and organ damage. It is my belief that this is preferred and safer as a research question and conclusion based on your analyses. So, I would also like the research question to be "to assess nutritional risk according to SLE activity, and explain active SLE via nutritional risk indexes."

We appreciate these useful comments. As suggested, we have modified the research question (see lines 96,97) and final conclusions (see line 296) in the revised version of the manuscript.

-References:

Ref 11 has issues.

We have checked and corrected reference 11.

Reviewer 2 Report

  The authors performed cross-sectional study on stable SLE patients that test the relationship between disease activity measures and nutritional scores. While their analysis is generally well-performed, several questions exist in the validity of their statistical analysis.

#1. Active SLE was defined based on SLEDAI-2K score score in this study. In 41 active SLE patients, affected organs (nephritis, arthritis, leukopenia etc) showed be shown.  

#2. SLEDAI-2K score includes DNA antibody, low complement, and WBC/platelet count in its calculation. Is it statistically sound to include ds-DNA antibody, complement or CBC count as explanatory variables in Table 3 linear regression model or Table 4 Logistic regression model?

#3. Also, the problem of multicollinearity showed be discussed in the manuscript.

#4. In their logistic regression model (Table 4), how did the authors choose variables?

Does inclusion of variables that are significant in univariate analysis only, or stepwise methods change the results?

#5. Table 3. The analysis of medical treatment should be described in more detail in methods section.

#6. line 76. The report by Ahn SS et al. should be cited correctly.

#7. line 146. (Fig 2b), should be omitted.

#8. line 218. Complement C4 OR should be corrected to 0.959.

Author Response

REVIEWER #2

-The authors performed cross-sectional study on stable SLE patients that test the relationship between disease activity measures and nutritional scores. While their analysis is generally well-performed, several questions exist in the validity of their statistical analysis.

-  #1. Active SLE was defined based on SLEDAI-2K score in this study. In 41 active SLE patients, affected organs (nephritis, arthritis, leukopenia etc) must be shown.  

Thanks for this important suggestion. We have added a few lines regarding this fact in the results section in the revised version of the manuscript (see lines 204-206).

-#2. SLEDAI-2K score includes DNA antibody, low complement, and WBC/platelet count in its calculation. Is it statistically sound to include ds-DNA antibody, complement or CBC count as explanatory variables in Table 3 linear regression model or Table 4 Logistic regression model.

We agree with the reviewer that it might not be statistically sound to include these variables, considering that SLEDAI-2K include them, as part of its calculation. However, we considered interesting to show them in the regression analysis since in a previous study performed in SLE by Ahn, S. et al. these variables were shown. Thus, we found of interest to provide this information to facilitate the scientific community the comparison of our and previous findings.

-#3. Also, the problem of multicollinearity showed be discussed in the manuscript.

The variance inflation factor (VIF), which quantifies the extent of correlation between one predictor and the other predictors in a model, was used for diagnosing multicollinearity. All the VIF obtained were near 1. Since a value of 1 means that the predictor is not correlated with other variables, we can conclude that we have no problem of multicollinearity in the regression analysis. In the statistical analysis section we have stated that VIF was used for diagnosing multicollinearity (see line 188).

-#4. In their logistic regression model (Table 4), how did the authors choose variables? Does inclusion of variables that are significant in univariate analysis only, or stepwise methods change the results?

We would like to note we have chosen the covariables based on previous studies. Previous evidence showed that the following variables: age, sex and medical treatment (immunosuppressants, antimalarials and/or corticosteroids) could influence SLEDAI and SLICC. Therefore, we have considered to include all of them.

-#5. Table 3. The analysis of medical treatment should be described in more detail in methods section.

As suggested also by reviewer #1 we have given further details about medical treatment (type of immunosuppressors) in the methods section (lines 184,185).

 -#6. line 76. The report by Ahn SS et al. should be cited correctly.

Thanks for noticing this mistake. As suggested also by reviewer#1 we have fixed this issue (see line 82).

-#7. line 146. (Fig 2b), should be omitted.

Thanks for noticing this. We have fixed it in the revised version of the manuscript.

-#8. line 218. Complement C4 OR should be corrected to 0.959.

Thanks for noticing this. We have fixed it in the revised version of the manuscript (see line 249).

Round 2

Reviewer 2 Report

Thank you for your reply on questions of the statistical analysis. 

The answers are reasonable and  understandable.    

The revised manuscript is now suitable for publication.